# Epicardial Adipose Tissue—A Novel Therapeutic Target in Obesity Cardiomyopathy

**DOI:** 10.3390/ijms26167963

**Published:** 2025-08-18

**Authors:** Kacper Wiszniewski, Anna Grudniewska, Ilona Szabłowska-Gadomska, Ewa Pilichowska-Paszkiet, Beata Zaborska, Wojciech Zgliczyński, Piotr Dudek, Wojciech Bik, Marcin Sota, Beata Mrozikiewicz-Rakowska

**Affiliations:** 1Department of Endocrinology, Centre of Postgraduate Medical Education, Marymoncka St. 99/103, 01-813 Warsaw, Poland; wiszniek@bielanski.med.pl (K.W.);; 2Lux Med Medical Centre in Warsaw, 49 Komitetu Obrony Robotników Street, 02-146 Warsaw, Poland; 3Laboratory for Cell Research and Application, Medical University of Warsaw, 02-097 Warsaw, Poland; 4Department of Cardiology, Centre of Postgraduate Medical Education, Grochowski Hospital, Grenadierów 51/59, 04-073 Warsaw, Poland; 5Department of Neuroendocrinology, Centre of Postgraduate Medical Education, Marymoncka St. 99/103, 01-813 Warsaw, Poland; 6Faculty of Medicine, Medical University of Warsaw, 02-091 Warsaw, Poland

**Keywords:** obesity, tirzepatide, GLP-1/GIP receptor agonists, heart failure, cardiomyopathy, epicardial adipose tissue, adipose tissue dysfunction

## Abstract

Obesity is strongly associated with an increased risk of heart failure. Recent studies indicate that epicardial adipose tissue plays a critical role in the development of obesity-related cardiomyopathy. This distinct visceral fat depot, located between the myocardium and the visceral pericardium, is involved in direct cross-talk with the adjacent myocardium, influencing both its structural integrity and electrophysiological function. This review aims to provide an up-to-date overview of the morphological, metabolic, immunological, and functional alterations of this adipose compartment in the context of obesity, and to explore its contribution to the pathogenesis of heart failure. Moreover, the article synthesizes current evidence on the potential cardioprotective effects of emerging anti-obesity pharmacotherapies—particularly GLP-1 and dual GLP-1/GIP receptor agonists—on metabolic pathways associated with epicardial fat that are implicated in obesity-induced cardiomyopathy. Further clinical trials are required to clarify the impact of these therapies on the course and prognosis of heart failure, as well as on the epidemiology and societal burden of the disease.

## 1. Introduction

Obesity is strongly associated with the risk of developing heart failure [1], both by directly affecting myocardial function and structure [2,3], as well as by increasing the likelihood of cardiovascular risk factors such as coronary artery disease, atrial fibrillation, type 2 diabetes mellitus, stroke, metabolic-associated fatty liver disease, obstructive sleep apnea, and mood or anxiety disorders (Figure 1) [4,5,6,7,8,9,10,11].

The pathomechanism of obesity-related cardiomyopathy is complex and includes myocardial remodeling with the development of systolic and diastolic dysfunction, increased circulating blood volume, cardiac output and vascular resistance in systemic and pulmonary circulation, decreased arterial compliance with metabolic dysregulation of the endothelium and oxidative stress, hormonal dysregulation involving increased activity of the renin–angiotensin–aldosterone system and hyperinsulinemia, as well as alterations in cardiomyocyte metabolism [2,3,12]. Moreover, it is believed that adipose tissue dysfunction and systemic inflammation are among the key drivers of the above alterations in cardiovascular function [13,14]. Under physiological conditions, the main reservoir of fat in humans is subcutaneous tissue, which protects other organs from lipotoxicity. However, continuous excessive intake of nutrients and positive energy balance exert anabolic stress on adipocytes, which initiates an inflammatory response, makes the subcutaneous tissue unable to store more lipids, and leads to visceral fat accumulation and ectopic fat deposition [15,16]. This process is truly detrimental, as visceral (VAT) and subcutaneous adipose tissue (SAT) are functionally and metabolically different due to unique secretomes which alter paracrine and endocrine signaling [17,18]. Unlike the SAT [17,19], VAT is associated with numerous cardiovascular risk factors including hypertension, dyslipidemia, diabetes/prediabetes, coronary artery disease, cardiac remodeling, and thus worse prognosis [20,21,22,23].

Epicardial adipose tissue (EAT) is a special type of VAT located between the myocardium and visceral pericardium that can directly affect the heart and coronary vessels due to its functional proximity and common microcirculation as well as its unique transcriptome [24,25]. Furthermore, the lack of anatomical barriers enables paracrine interaction and direct crosstalk between EAT and the contiguous myocardium [26,27]. Several studies have shown that EAT accumulation is associated with a number of factors leading to the development of heart failure, including ventricular hypertrophy, increased cardiac filling pressures, and myocardial fibrosis [28,29,30,31]. On the other hand, recent studies have shown that EAT function is modulated by incretin axis hormones, which contribute to the regulation of fatty acid metabolism, adipogenesis, epicardial fat inflammation [32,33,34,35]. Given the above, EAT is an attractive therapeutic target for the treatment of obesity-related cardiomyopathy involving incretin mimetics.

## 2. EAT Function Under Physiological Conditions

EAT is a one-of-a-kind fat deposit of mesodermal origin located between the myocardium and the visceral pericardium, which contains adipocytes, ganglia, and nerves, as well as inflammatory, immune, and stromal cells [36].

EAT is directly adjacent to the myocardium and covers 80% of its surface in healthy individuals, although in the disease state it may also extend between muscle fibers, resulting in cardiac steatosis and arrhythmogenesis [37,38]. Therefore, EAT provides mechanical support for the underlying myocardium and is considered to shield coronary arteries from tension as well as excessive distortion during cardiac systole due to its elasticity [39].

EAT demonstrates distinct histological properties, as epicardial adipocytes are smaller compared to those in subcutaneous fat [40]. Interestingly, Bambace et al. showed that adipocyte size is dependent on insulin resistance as well as serum leptin and adiponectin concentrations [40]. Despite the fact that EAT is considered white adipose tissue, it has a unique lipidomic profile. Barchuk et al. showed that compared to SAT, EAT contains more phosphatidylcholine, sphingolipids, phosphatidylethanolamine, and phosphatidylethanolamine plasmalogens, as well as more short- to medium-chain saturated fatty acids [41]. This lipidomic profile is closely related to EAT function. Short- and medium-chain saturated fatty acids are an efficient substrate for mitochondrial beta-oxidation, which can either flow into the coronary bloodstream or diffuse through the interstitial fluid and serve as an excellent source of energy for the underlying myocardium, especially during periods of high energy requirement [38]. On the other hand, the capacity of EAT to rapidly metabolize free fatty acids prevents cardiotoxicity caused by their possible excess [38]. Moreover, a recent study by Park et al. [42] showed that plasmalogens regulate cold-induced adipocyte thermogenesis by mediating mitochondrial fission. In addition, Sacks et al. showed that EAT contains UCP-1, which is a proton channel typically present in the inner membrane of brown adipose tissue mitochondria that dissipates the electrochemical gradient produced by oxidative phosphorylation and thus generates heat [43]. This study further revealed that compared to SAT, EAT demonstrates a greater expression of genes associated with brown adipose tissue development and/or white adipose tissue browning [43]. Hence, EAT exhibits some “beige” features and is considered to be involved in thermogenesis.

EAT receives blood supply from branches of the coronary arteries, which enables direct paracrine and vasocrine crosstalk with the myocardium via a common microcirculation (Table 1) [44,45]. Compared to the SAT, EAT contains significantly higher concentrations of pro-inflammatory cytokines, including IL-1β, IL-6, MCP-1, and TNF-α [46,47]. Interestingly, this property is independent of body mass index [46]. In addition, Shaihov-Taper et al. proved that such pro-inflammatory and profibrotic molecules can be secreted by the EAT while encapsulated within extracellular vesicles [48]. Consistently, Schleinitz et al. [48] proved that compared to SAT, EAT demonstrates a greater expression of genes involved in inflammation, extracellular matrix remodeling, thrombosis, coagulation, apoptosis and beigeing such as GATA4, CD68, COL4A4, and HBM. On the other hand, EAT may exhibit anti-inflammatory, anti-fibrotic, and anti-atherogenic properties as well [38,47]. Acting via cytokines such as adiponectin, adrenomedullin, and omentin, it exerts protective effects on the endothelium and reduces oxidative stress [38,47].

## 3. How Does EAT Contribute to the Development of Heart Failure in Obesity?

### 3.1. Cytokines Secretion and Immune Cells Activation

Like all kinds of white adipose tissue, EAT is poorly vascularized. Moreover, in obese patients, blood flow through this tissue does not increase in proportion to its accumulation (Figure 2) [47]. Therefore, as obesity results in an increase in adipocyte size that exceeds the maximum distance of efficient intratissue oxygen diffusion, local hypoxia develops [48,49]. The hypoxemic environment induces oxidative stress and impairs numerous metabolic processes, including folding of proteins synthesized in the endoplasmic reticulum, insulin signaling, Akt phosphorylation, and mitochondrial ATP generation [50,51,52]. In response to these conditions, cells in the EAT activate hypoxia-inducible factor (HIF), which is a family of transcription factors that induce metabolic pathways involved in adaptation to hypoxia [53,54]. As a consequence, adipose tissue homeostasis is disrupted, as its metabolic and endocrine functions and the profile of secreted cytokines are shifted towards a pro-inflammatory and pro-fibrotic state [50]. EAT dysregulation in obese patients thus includes increased secretion of TNF-α, IL-6, IL-1β, leptin and resistin, as well as reduced production of the anti-inflammatory adiponectin [27,55,56]. In addition, EAT releases MCP-1, MIP, IFNγ and leukotriene B4, which catalyse the infiltration of macrophages and their transition from M2 to the inflammation-promoting M1 phenotype [50,51,52]. Recruitment of monocytes from the blood is facilitated by an upregulated local concentration of IL-6, leptin, resistin, and MCP-1, which increase the expression of vascular endothelial adhesion molecules [51]. Infiltration of immune cells in expanding EAT further modulates the inflammatory microenvironment. For example, M1 macrophages secrete pro-inflammatory cytokines involving TNF-α, IL-6, IL-1β, MCP-1, and plasminogen activator inhibitor 1 (PAI-1), which induce reciprocal macrophage activation and promote the M1 phenotype, resulting in a vicious cycle [51]. The aforementioned inflammatory mediators have a direct impact on heart failure pathogenesis and progression via immunological cross-talk between cardiomyocytes, adipocytes, and fibroblasts, as well as epithelial and immune cells.

TNF-α contributes to cardiac remodeling and impairs both systolic and diastolic function through uncoupling of beta-adrenergic receptors and a reduction in intracellular calcium concentration during systole [57,58]. Calcium dysregulation due to TNF-α results from the downregulation of sarcoplasmic reticulum calcium ATP-ase (SERCA) and phospholamban [57,58]. In addition, TNF-α promotes cardiac hypertrophy and fibrosis, which is mediated by cardiac fibroblast proliferation, transition of fibroblasts into pathogenic myofibroblasts, fibronectin deposition, activation of metalloproteinases, and inhibition of metalloproteinase inhibitors [57,59,60,61,62]. A study by Ahmed et al. revealed that TNF-α-dependent metalloproteinase activation occurs via the PI3Kγ signaling pathway [63]. TNF-α may also induce inflammation, tissue remodeling, and apoptosis via nuclear factor kappa B (NFκB) [64], although TNF-α-mediated apoptosis induction is also possible via RIP1–RIP3–MLKL axis activation by apoptosis signal-regulating kinase 1 (ASK1) [65,66]. TNF-α potentiates the inflammatory cascade through the mitogen-activated protein kinase (MAPK)/activator protein-1 (AP-1) signaling pathways by promoting the secretion of monocyte chemoattractant proteins (MCP)-1 and MCP-3, which not only drive monocyte recruitment but also reversibly increase TNF-α expression [67,68]. Within endothelial cells, TNF-α stimulates the expression of adhesion molecules, increases microvascular permeability, and decreases nitric oxide production, as well as enhancing oxidative stress through the activation of NADH oxidase [66,69]. These excess reactive oxygen species damage mitochondrial DNA in cardiomyocytes via the sphingomyelin–ceramide signaling pathway [70,71]. Furthermore, TNF-α promotes LDL transcytosis as well as macrophage and smooth muscle cell proliferation, thus contributing to foam cell formation and the development of atherosclerosis [72,73,74,75].

IL-1 is another pro-inflammatory cytokine significant in heart failure pathogenesis [58,76], as it impairs beta receptor stimulation-related activation of adenylyl cyclase responsible for cAMP synthesis, reduces intracellular calcium ion influx through L-type channels, and decreases calcium reuptake in the sarcoplasmic reticulum via downregulation of calcium-ATPase protein and mRNA level [77,78]. In addition, according to Zell et al., both IL-1 and TNF-α reduce energy production and myocardial contractility by inhibiting pyruvate dehydrogenase activity and mitochondrial function [79]. Moreover, IL-1 signaling mediated by phosphoinositide-3-kinase-γ contributes to the reduction in phosphodiesterase-3 activity and myocardial contractility as well as leading to desensitization of beta adrenergic receptors [80]. The beneficial effects of IL-1 blockade therapy in heart failure provide indirect experimental confirmation of the detrimental role of IL-1. Moroni et al. showed that in Black African American patients with heart failure, anakinra therapy leads to a decrease in blood C-reactive protein levels and intracardiac pressures as well as improved quality of life and ventilatory efficiency [81].

IL-6 also exerts pleiotropic effects on both the myocardium and the vascular system [82,83]. Within the myocardium, IL-6 enhances TGF-β1-mediated MMP2/MMP3 signaling, which induces myofibroblastic proliferation and fibrosis [84]. The effect of IL-6 on myocardial fibrosis depends on the activation of the MAPK and CAMKII-STAT3 pathways [83]. In line with the above, recent studies by Berger et al. and Alogna et al. showed that elevated IL-6 levels in heart failure patients with preserved ejection fraction correlate with more upper body fat accumulation, greater symptom severity, and increased mortality [85,86]. Moreover, IL-6, via several signaling pathways including the JAK/STAT pathway, enhances vascular inflammation by promoting endothelial dysfunction as well as smooth muscle cell proliferation and migration, which leads to atherosclerotic plaque formation and destabilization, thus contributing to ischemic cardiomyopathy [83,87,88,89].

Interestingly, pro-inflammatory cytokines secreted by EAT can affect the electrical conduction properties of cardiomyocytes. IL-6 promotes L-type calcium channels while inhibiting fast and slow potassium rectifying currents as well as large inward sodium currents, which overall increases calcium load in the myocytes and prolongs the action potential (AP) duration as well as the QT interval [1,2,3,4,5,6,7,8,9,90]. Mechanistically, IL-6 signaling modifies ventricular ion channel function by altering their gating properties, impairing the trafficking of the channel protein to the cell surface, or in a combination of the above [90]. Furthermore, IL-1 has been shown to similarly promote L-type calcium channels, while TNF-alpha inhibits potassium rectifying currents [91,92]. Altogether, this implies that cytokines contribute to the electrical instability of the myocardium and increased risk of ventricular arrhythmias in obese patients.

The effect of leptin on the myocardium is complex and controversial. Transient exposure to leptin results in the inhibition of fatty acid oxidation by cardiomyocytes in favor of glucose utilization, which increases the efficiency of energy production while reducing myocardial oxygen demand and lipid accumulation along with associated lipotoxicity. These acute effects of leptin appear to provide cardioprotection during stress conditions such as ischemia [93,94]. On the other hand, chronic exposure to leptin is detrimental for cardiovascular health, which may be related to the development of tissue resistance to its initial protective effects [94,95,96]. Animal model studies revealed that sustained leptin signaling promotes fibroblasts proliferation, cardiac fibrosis, and cardiomyocyte hypertrophy. These pathological alterations are mediated by the production of transforming growth factor-beta (TGF-β), galectin-3, and connective tissue growth factor production (CTGF) via the activation of the mechanistic target of the rapamycin (mTOR) pathway [97]. Moreover, leptin increases the synthesis of endothelin-1 (ET-1), which exerts direct pro-hypertrophic and pro-fibrotic effects on cardiomyocytes and cardiac fibroblasts. This process is augmented by leptin-dependent synthesis of reactive oxygen species (ROS), which further upregulates ET-1 production [94]. The ET-1 signaling pathway as well as increased oxidative stress also appear to impair myocardial lusitropy by disrupting calcium homeostasis, primarily through reduced SERCA expression and increased phospholamban levels. This imbalance hinders calcium reuptake into the sarcoplasmic reticulum, leading to delayed myocardial relaxation and diastolic dysfunction. Leptin further indirectly contributes to myocardial hypertrophy by stimulating central sympathetic activity, which elevates arterial blood pressure and heart rate, leading to an increased afterload and additional hemodynamic stress on the myocardium [98,99]. In addition, persistent hyperleptinemia may stimulate vascular smooth muscle cells to proliferate and undergo calcification as well as impair endothelial function by suppressing the expression of peroxisome proliferator-activated receptor gamma (PPAR-γ), a critical factor in facilitating nitric oxide (NO)-mediated vasodilation. Thus, leptin contributes to increased arterial rigidity, progression of atherosclerosis, and ischemic cardiomyopathy [100,101,102]. Moreover, leptin stimulates the secretion of pro-inflammatory cytokines including TNF-α and IL-6, exacerbating their detrimental effect on cardiac function.

A recent study by Aztatzi-Aguilar revealed that oxidative stress conditions in the dysfunctional adipose tissue increase the concentration of osteopontin (OPN) [103], which is known to be associated with tissue remodeling, fibrosis, and inflammation, as well as diabetes and cardiovascular complications [103,104,105]. In line with this study, Luna-Luna et al. confirmed that OPN synthesis occurs in EAT as well, which correlates with the development of coronary artery disease [106]. OPN increases vascular resistance as well as inducing vascular calcification and atherosclerotic plaque formation [107,108,109]. Moreover, Strobescu-Ciobanu et al. revealed that OPN expression in atherosclerotic plaques of patients who underwent carotid endarterectomy enhanced plaque ulceration and instability [110]. Another study by Yousefi et al. revealed that in a mouse model of heart failure induced by Col4a3 deficiency, lack of OPN resulted in reduced cardiac hypertrophy and improved diastolic function [111]. Furthermore, OPN has been shown to promote pathological cardiac remodeling and fibrosis through increased collagen production and extracellular matrix deposition [112,113,114]. An animal model study by Herum et al. demonstrated that this phenomenon may be related to an increase in the expression and activity of lysine oxidase, which is responsible for the synthesis of insoluble cross-linked collagen [115,116]. It is worth noting that OPN-induced myocardial fibrosis escalates the risk of developing atrial fibrillation [117,118]. Myocardial fibrosis is intensified by the recruitment of inflammatory cells, which is also stimulated by OPN [64,113,119]. In line with this study, Sojeima et al. showed that the level of circulating CD 4+ lymphocytes expressing OPN is correlated with New York Heart Association (NYHA) class in patients with heart failure [119]. Moreover, OPN has been proven to enhance macrophage differentiation and local proliferation [120], whereas, on the other hand, IL-10 and M-CSF act synergistically to stimulate OPN expression in cardiac macrophages through the mediation of STAT3 and ERK signaling pathways [121,122]. Additionally, studies on animal models revealed that OPN overexpression is associated with increased cardiomyocyte apoptosis [123]. Through the above pathophysiological pathways, OPN secreted by EAT contributes to the development of heart failure, while its levels correlate with the risk of hospitalization due to exacerbation of the disease [108].

### 3.2. Impaired Ventricular Compliance Due to Abnormal Intracardiac Mechanical Interplay

Like any cardiac chamber, the pericardium exhibits a distinct curvilinear pressure–volume relationship. In obese patients, the accumulation of EAT within the anatomically restricted pericardial space contributes to the development of heart failure by augmented pericardial restraint and intensified biventricular mechanical coupling [28,31,124]. The presence of pericardial restraint is reflected by higher intrapericardial pressures both at rest and during exercise, in contrast to non-obese patients [124]. As a result, intraventricular pressures begin to equilibrate, the interventricular septum flattens and shifts less toward the right ventricle, and mechanical independence between the ventricles diminishes [124,125]. This leads to the disruption of the physiological relationship between left ventricular filling pressure and preload. Consequently, elevated left ventricular filling pressures cause an increase in pulmonary capillary hydrostatic pressure, contributing to dyspnea, particularly during exertion [124]. These relationships are reflected by a study by Min et al. [126], which demonstrated an association between EAT and abnormalities in right ventricular and atrial strain, as well as reduced exercise capacity measured by the six-minute walk test, further suggesting a potential link with the development of heart failure.

### 3.3. EAT and Arrhythmogenesis

Abnormal impulse generation associated with pacemaking and triggered activity, as well as abnormal impulse conduction, is traditionally considered the two main mechanisms of cardiac arrhythmia. EAT alters cardiac electrophysiology due to adipocyte infiltration into the myocardium (Figure 3). Adipocytes have been observed to form narrow, densely arranged strands extending from the epicardium and infiltrating the myocardium between cardiomyocyte bundles [127]. Alternatively, they may be encapsulated within regions of dense fibrosis, consistent with a scar-like morphology indicative of fibro-fatty infiltration [127]. Emerging evidence indicates that these adipose deposits may undergo structural remodeling in response to inflammatory stimuli, contributing to the progression of fibro-fatty tissue replacement [128].

It is important to emphasize that fibrous fatty replacement is not a process specific to obesity-related cardiomyopathy. Although it may occur in various cardiac disorders, it is classically observed in arrhythmogenic right ventricular cardiomyopathy (ARVC), which may suggest a potential overlap in certain pathogenic mechanisms underlying these two conditions [129]. These diseases, however, differ fundamentally in their molecular basis. In obesity-related cardiomyopathy, fatty-fibrous infiltration is primarily driven by metabolic overload and cytokine-mediated injury, leading to cellular dysfunction, extracellular matrix remodeling, and epithelial-to-mesenchymal transition of epicardial-derived cells [128]. In contrast, in ARVC, this process is most commonly associated with mutations in genes encoding desmosomal proteins, and less frequently with mutations in genes involved in adherens junctions, cytoskeletal structures, or ion channels and their modulators [129]. Although both types of cardiomyopathy involve structural remodeling of the myocardium and may progress to overt heart failure, differences in their molecular pathogenesis account for distinct clinical presentation and markedly increased risk of life-threatening arrhythmias observed in ARVC [129]. Despite recent advancements, the biochemical mechanisms of ARVC remain incompletely understood. Interestingly, the application of spatial transcriptomics has recently revealed that in patients with a pathogenic desmoplakin variant, an increase in endothelial PAS domain-containing protein 1 (EPAS1) in stressed cardiomyocytes correlates with mitochondrial dysfunction, increased apoptosis, hypoxemic stress, and impaired contractility [130]. Moreover, Boogerd et al. demonstrated that in patients with ARVC, the transcription factor ZBTB11, when subjected to external stimuli, may act as a potential inducer of cardiomyocyte atrophy, apoptosis, and remodeling [131]. Using spatial transcriptomics, they identified ZBTB11 as being enriched in cardiomyocytes adjacent to fibro-fatty infiltrates in ARVC. Nevertheless, further studies are needed to elucidate whether ZBTB11 or EPAS1 play a similar role in other forms of cardiomyopathy, including obesity-related cardiomyopathy.

Regardless of its origin, adipocyte infiltration creates an anatomical barrier between cardiomyocytes, leading to nonuniform anisotrope propagation of the AP wave within the myocardium, which results in conduction delay and contributes to re-entry arrhythmias [132,133]. Nalliah et al. showed that in patients with coronary artery disease, EAT volume is associated with local fibrosis, slower conduction, and lateralization of atrial gap-junction protein connexin 40 [134].

**Figure 3 ijms-26-07963-f003:**
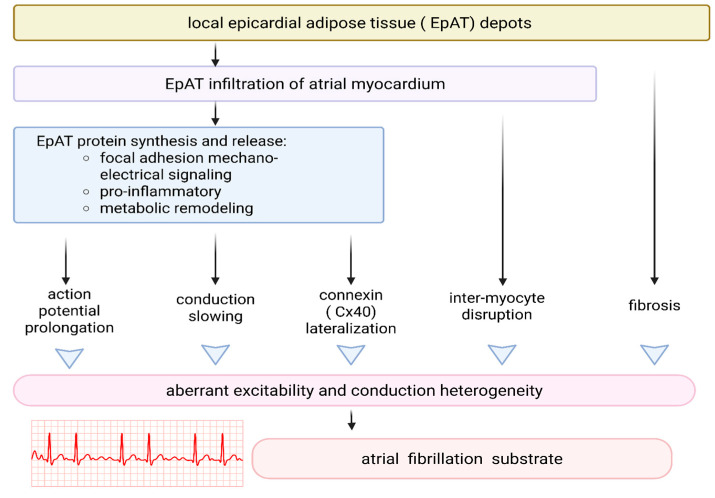
Local adipose tissue infiltration on atrial myocardium and its influence on atrial conduction [131]. Created in BioRender. Grudniewska, A. (2025) https://BioRender.com/wh2ifi6 (accessed on 17 August 2025).

Moreover, it has been shown that some cells contained in EAT such as myofibroblasts and macrophages can form gap junctions with cardiomyocytes, although it is not known whether the same is also true for adipocytes [135,136,137]. Given that the resting potential of adipocytes, macrophages, and fibroblasts is significantly higher than that of cardiomyocytes (−10 to −30 mV vs. −90 mV), the presence of gap junctions between these cells may cause direct exchange of ions and current [135,138]. Slight depolarization facilitates myocardial excitability, as it brings the resting membrane potential closer to the activation threshold [133]. As depolarization progresses, partial or complete inactivation of voltage-gated fast sodium channels occurs, resulting in a consequent reduction in conduction velocity [133]. A computer modeling study by MacCannell et al. revealed that fibroblast–cardiomyocyte coupling affects the shape of the AP, shortening its duration and reducing its plateau level [139]. In line with this study, Kostecki et al. demonstrated that myofibroblasts co-cultured with cardiomyocytes induce abnormal automobility as well as contribute to AP shortening and a reduction in conduction velocity [140]. It can be hypothesized that gap junctions between adipocytes and cardiomyocytes may exert similar electrophysiological properties [133].

Furthermore, cardiomyocyte electrophysiology is directly modulated by the EAT secretome [80,141]. A novel study by Ernault et al. demonstrated that EAT secretes extracellular vesicles (EVs) containing miR-1-3p and miR-133a-3p, which, according to rat models, slow electrical conduction and downregulate the expression of inward-rectifier type potassium channels Kcnj2 and Kcnj12 [142]. Animal model studies indicate that pro-inflammatory cytokines, including IL-1β and TNF-α, promote myocardial electrical remodeling by attenuating the repolarizing transient outward potassium current, thereby prolonging AP duration [143,144]. Furthermore, Saraf et al. proved that in human induced pluripotent stem cell-derived cardiomyocytes, TNF-α elicits dose-dependent cytosolic calcium transients as well as calcium sparks—defined as calcium release events from cardiac sarcoplasmic reticulum—which may contribute to the initiation of arrhythmogenic activity [143]. According to this study, TNF-α has also been found to down-regulate the expression of several gap junction proteins including connexin 26, connexin 32, connexin 36, connexin 40, and connexin 43, thus altering AP propagation. Conversely, animal model studies demonstrated that the expression of connexin 40 and connexin 43 is further downregulated by IL-1β and IL-6 [82,83]. Moreover, IL-6 has been shown to modulate L-type calcium channel function as well as to downregulate both the expression and activity of SERCA2, potentially impacting AP duration and promoting calcium-mediated arrhythmic events [84].

## 4. Pharmacological Modulation of EAT Function—Potential Implications for the Pathogenesis of Obesity-Related Cardiomyopathy

Several classes of drugs are used to treat obesity, including GLP-1 receptor (GLP-1R) agonists, agonists of glucose-dependent insulinotropic polypeptide (GIP) and GLP-1 receptors, lipase inhibitors, sympathomimetics, and dopamine and norepinephrine reuptake inhibitors with the addition of an opioid antagonist, in particular bupropion/naltrexone [145]. Emerging evidence suggests that GLP-1 and dual GLP-1/GIP receptor agonists exert modulatory effects on both the volume and functional properties of epicardial adipose tissue (EAT), thereby potentially mitigating the risk of obesity-related cardiomyopathy. Conversely, current data do not support a direct impact of other pharmacological classes of anti-obesity agents on EAT, and any potential benefits are most plausibly secondary to generalized adiposity reduction.

### 4.1. Molecular and Functional Pathways Mediating the Effects of GLP-1 and GLP-1/GIP Receptor Agonists on EAT

Glucagon-like peptide-1 (GLP-1) and glucose-dependent insulinotropic polypeptide (GIP) are components of the incretin axis, which is essential for insulin sensitivity, appetite control, and glucose regulation. Therapy with GLP-1 or GLP-1/GIP receptor agonists is currently the first-line pharmacological treatment for obesity. Tirzepatide, liraglutide, and semaglutide induce satiety, decrease stomach emptying, suppress appetite, and improve insulin sensitivity and secretion [146]. These obesity medications have been shown to be crucial in lowering total body weight and fat levels, which also results in a reduction in EAT volume and lower cardiovascular risk. Moreover, GLP-1 and GLP-1/GIP receptor agonists exert direct effects on EAT (Figure 4). This finding is supported by Malvazos et al., who demonstrated that EAT contains receptors for both GLP-1 and GIP, mainly in macrophages and partially in adipocytes [32]. In addition, GLP-1R activation has been shown to stimulate brown adipose tissue thermogenesis and beneficial white-to-brown conversion of adipose tissue in murine models [147]. This observation was later confirmed by a study by Janssen et al., which demonstrated that during a 12-week therapy of exenatide in nondiabetic males, there was an increase in brown adipose tissue metabolic volume and fluorodeoxyglucose uptake by brown adipose tissue, without affecting oxidative resting energy [148]. Furthermore, according to a double-blind, placebo-controlled trial with 35 overweight and obese patients conducted by Corbin et al., GLP-1R activation is associated with enhanced fatty acid oxidation, which promotes EAT mass reduction [149]. In another study, Dozio et al. assessed EAT thickness and performed a microarray analysis of GLP-1R, GLP-2R, and genes involved in fatty acid metabolism in patients with coronary artery disease undergoing coronary artery bypass grafting. The researchers demonstrated that EAT GLP-1R expression is directly correlated with genes promoting beta-oxidation and differentiation of white-to-brown adipose tissue, while showing an inverse correlation with pro-adipogenic genes. On the other hand, GLP-2R expression is positively correlated with genes associated with adipogenesis and lipid synthesis, while being inversely correlated with genes promoting beta-oxidation. Notably, GLP-1 and GLP-2 levels were elevated in coronary artery disease patients compared to control and other patients with increased EAT thickness. These results indicate that GLP-1 analogs may directly target EAT GLP-1R, therefore reducing local adipogenesis and enhancing fat utilization, while simultaneously promoting adipose tissue browning. Given that EAT is in direct proximity to the myocardial arteries, the beneficial effects of GLP-1 activation may extend to the myocardium [34].

Moreover, GLP-1R agonists suppress inflammation induced by cytokines produced in epicardial adipose tissue, which directly contributes to reduced risk of myocardial dysfunction and following cardiomyopathy. This was shown by Von Scholtern et al., who demonstrated that liraglutide treatment results in a 12% reduction in TNF-α, a 4% reduction in the cardiovascular risk marker MR-proADM, and a 13% reduction in the heart failure risk marker MR-proANP [150]. In line with this study, Daousi et al. showed that GLP-1 infusion results in the suppression of IL-6 at 120 min and 180 min following drug administration, both in healthy and diabetic patients [151]. Anti-inflammatory properties of the GLP-1R agonists were summarized in a meta-analysis by Bray et al., which associated the treatment with significant reductions in CRP, TNF-α, and MDA, as well as with a significant increase in adiponectin [152]. The positive effect of GLP-1R agonists on adiponectin level was also confirmed by a meta-analysis of randomized controlled trials by Simental-Mendia et al. Additionally, Zobel et al. demonstrated that liraglutide induces a downregulation of TNF-α and IL-1β in peripheral blood mononuclear cells (PBMCs). However, the lack of evidence for GLP-1R’s presence on these cells suggests that this effect is indirect and does not act on PBMC gene transcription [153]. Furthermore, GLP-1R agonists have been shown to inhibit plaque progression as well as reduce plaque macrophage infiltration and calcification in hyperlipidemic rabbits, which suggests a protective effect of these drugs against ischemic myocardial injury [154].

A recent study by Chrostopher et al. indicates that cardioprotective effects of tirzepatide on cardiac remodeling are reflected in macroscopic morphological changes in both epicardial adipose tissue (EAT) and myocardial structure. The researchers demonstrated that 52-week tirzepatide therapy effectively reduced left ventricular mass by 11 g, as measured by cardiac magnetic resonance imaging, compared to placebo. Moreover, a significant decrease in EAT volume was observed. These findings suggest that tirzepatide may represent a potential therapeutic option for obesity-related heart failure with preserved ejection fraction [155].

### 4.2. Effects of Therapy with GLP-1 vs. Dual GLP-1/GIP Receptor Agonists

Both the direct action on EAT and the indirect effect through overall weight reduction suggest that GLP-1 receptor agonists and GIP analogs hold significant potential for exerting beneficial effects in cardiovascular disease. A meta-analysis by Berg et al. showed that 6-month-long GLP-1 receptor agonist therapy leads to a mean reduction in EAT thickness of 1.83 mm [156]. Additionally, a parallel study by Iacobellis et al. showed that after 6–12 months of treatment, semaglutide and dulaglutide reduce EAT thickness by approximately 20%, according to ultrasound measurements [157]. Moreover, a meta-analysis by Myasoedova et al. revealed that GLP-1R agonists caused a greater reduction in EAT than other cardiometabolic drugs such as SGLT2 inhibitors and statins [158]. While the direct effects of tirzepatide on EAT are not well studied, two meta-analyses by Alkhezi et al. and Permana et al. indicate that this dual GLP-1/GIP receptor agonist contributes to greater weight reduction, with a safety profile comparable to that of GLP-1 receptor agonists [159,160]. In general, the cardioprotective effects of GLP-1R agonists and tirzepatide are directly supported by findings from a recent meta-analysis by Stefanou et al., who found that these drugs reduce the risk of major adverse cardiovascular events (MACE) and all-cause mortality in patients with obesity [161]. Furthermore, evidence from the SUMMIT trial demonstrated that in obese patients with heart failure with preserved ejection fraction, tirzepatide reduces the risk of a composite endpoint involving heart failure exacerbation and cardiovascular death [162].

## 5. Conclusions and Future Directions

Although obesity-related cardiomyopathy and heart failure represent a significant public health issue, their complex pathogenesis remains incompletely understood. Recent studies increasingly highlight the detrimental role of morphological, functional, metabolic, and immunological transformation of EAT in the development of these conditions. This process is reflected in the EAT secretome, which, through direct paracrine and vasocrine crosstalk, contributes to structural remodeling of the underlying myocardium and affects its electrophysiological properties. Through both direct and indirect actions on multiple aspects of the detrimental transformation of EAT, GLP-1 and dual GLP-1/GIP receptor agonists offer a novel approach to obesity treatment that may influence metabolic pathways leading to obesity-related cardiomyopathy.

However, the molecular basis underlying the beneficial cardioprotective effects of this drug class remains to be fully elucidated. Robust evidence based on epicardial adipose tissue (EAT) biopsies is still lacking, particularly regarding the impact of GLP-1 and dual GLP-1/GIP receptor agonists on the local microenvironment, adipocyte phenotype, cytokine secretion profile, and immune cell activation. Clarifying the impact of this therapy on myocardial ion channel regulation and action potential propagation remains an important objective, particularly in the context of potential effects of treatment on arrhythmogenic risk. Moreover, the application of advanced imaging techniques, such as T1- and T2-weighted magnetic resonance mapping or myocardial strain analysis, appears to be a promising approach for the in vivo assessment of therapy-induced alterations in myocardial microstructure and function, including fibrofatty infiltration.

We hope that the promising data regarding the beneficial effects of GLP-1 and dual GLP-1/GIP receptor agonists on EAT volume and function will not only be confirmed but also complemented in future clinical trials, providing further insight into their impact on the course and prognosis of obesity-related cardiomyopathy. An open question remains as to whether, and to what extent, the increasingly widespread use of this drug class will ultimately influence the epidemiology and societal burden of heart failure as well as its complications. 

## Figures and Tables

**Figure 1 ijms-26-07963-f001:**
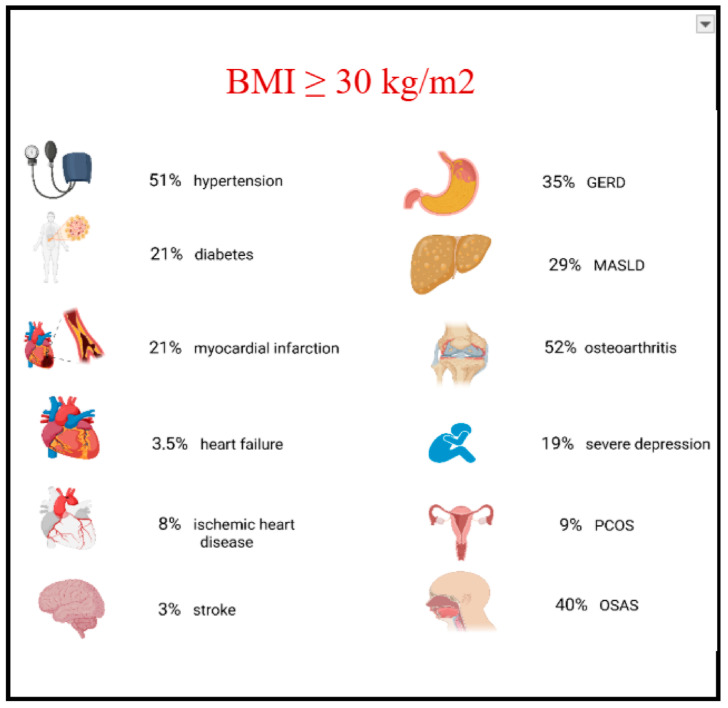
Prevalence of selected complications in people with obesity-related disease. Created in Biorender. Grudniewska, A. (2025) https://BioRender.com/in262l0 (accessed on 17 August 2025). Abbreviations: GERD, gastroesophageal reflux; MASLD metabolic associated fatty liver disease; OSAS, obstructive sleep apnea syndrome; PCOS, polycystic ovary syndrome; BMI, body mass index.

**Figure 2 ijms-26-07963-f002:**
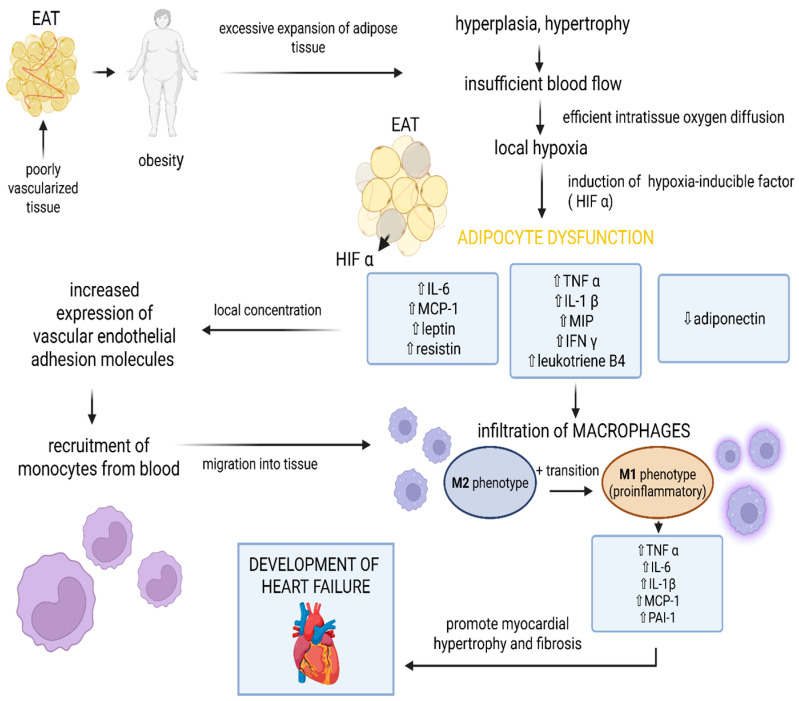
The impact of EAT on the development of heart failure in obese patients. Created in BioRender. Grudniewska, A. (2025) https://BioRender.com/a01nqmf (accessed on 17 August 2025). Abbreviations: EAT, epicardial adipose tissue; HIV, hypoxia-inducible factor; IL, interleukin; MCP-1, monocyte chemoattractant protein-1, TNF α, tumor necrosis factor-alpha; MIP, macrophage inflammatory protein, IFN γ, interferon γ; PAI-1, Plasminogen activator inhibitor-1.

**Figure 4 ijms-26-07963-f004:**
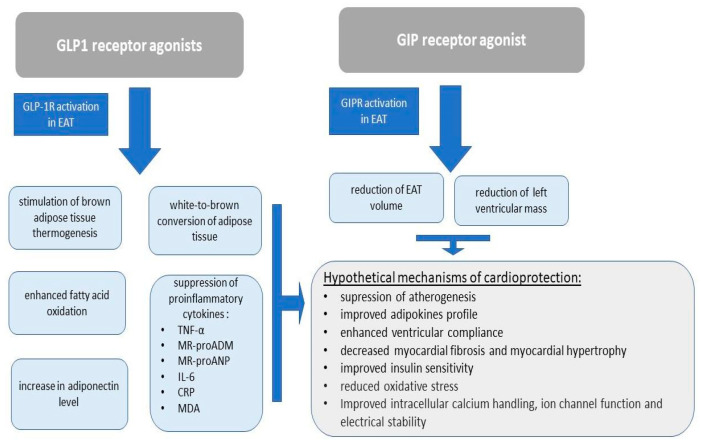
Molecular and functional pathways mediating the effects of GLP-1 and GLP-1/GIP receptor agonists on EAT as well as possible mechanisms contributing to cardioprotection.

**Table 1 ijms-26-07963-t001:** Functional properties of EAT [36].

Function	Cytokines Related with Particular Function
Pro-inflammatory, proatherogenic	TNF-α
	MCP-1
	IL-1, IL-1β, IL-1Ra, IL6, IL8, IL10 CRP PAI-1 Prostaglandin D(2), haptoglobin, α1-glycoprotein, JNK sPLA2-IIA, fatty-acid-binding protein 4 RANTES ICAM
Proliferative factors	NGF
	FLT-1
Anti-inflammatory, anti-atherogenic	Adiponectin Adrenomedullin
Insulin mimetic, markers of visceral fat	Resistin
	Visfatin Omentin
Brown fat differentiation transcription factors	PRDM-16
	PGC-1α
Thermogenic	UCP-1
Vascular remodeling, blood pressure control, myocardial hypertrophy, adipogenesis	Angiotensin Angiotensinogen Leptin
Receptors	Angiotensinogen type 1 receptor
	TLRs PPAR-γ GLUT-4

Abbreviations: CRP, C-reactive protein; FLT-1, soluble vascular endothelial growth factor receptor; GLUT-4, glucose transporter-4; ICAM, soluble intercellular adhesion molecule; IL, interleukin; IL-1Ra, interleukin-1 receptor antagonist; JNK, c-Jun N-terminal kinase; MCP-1, monocyte chemoattractant protein-1; NGF, nerve growth factor; PAI-1, plasminogen activator inhibitor-1; PGC-1α, PPAR-γ coactivator-1α; sPLA2-IIA, secretory type II phospholipase A2; PPAR-γ, peroxisome-proliferator-activated receptor γ; PRDM-16, brown adipocyte differentiation transcription factor PR-domain-missing16; RANTES, regulated upon activation normal T cell and secreted; TLRs, toll-like receptors; TNF-α, tumor necrosis factor-alpha; UCP-1, uncoupling protein-1.

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
