# Peer review of "Epicardial Adipose Tissue—A Novel Therapeutic Target in Obesity Cardiomyopathy"

_ijms, 2025, doi:10.3390/ijms26167963_

Round 1

Reviewer 1 Report

Comments and Suggestions for Authors

In the review article 'Epicardial adipose tissue- a novel therapeutic target in obesity cardiomyopathy' submitted by Wiszniewski et al to IJMS, the authors summarize the knowledge about epicardial adipose tissue and signalling pathways involved. The topic of this review article is interesting and the manuscript has a clear very convincing structure. So I have only some small minor points, which can be addressed in a minor revision.

1). I am wondering why the authors do not discuss fibro-fatty replacement, which is frequently found in patients with ARVC. Recently, spatial transcriptomic analysis revealed interesting insights into this replacement (see 'Spatial transcriptomics unveils ZBTB11 as a regulator of cardiomyocyte degeneration in arrhythmogenic cardiomyopathy' and 'EPAS1 induction drives myocardial degeneration in desmoplakin-cardiomyopathy').

2.) Therefore, I suggest to discuss also similarities and differences between this fat-replacement in obesity-related cardiomyopathy and ARVC (which is caused by mutations in genes, encoding desmosomal cell-cell adhesion proteins).

3.) It would be great if the authors can increase the size of their nice figures. 

In summary I suggest a minor revision.

Author Response

1). I am wondering why the authors do not discuss fibro-fatty replacement, which is frequently found in patients with ARVC. Recently, spatial transcriptomic analysis revealed interesting insights into this replacement (see 'Spatial transcriptomics unveils ZBTB11 as a regulator of cardiomyocyte degeneration in arrhythmogenic cardiomyopathy' and 'EPAS1 induction drives myocardial degeneration in desmoplakin-cardiomyopathy').

Thank you for drawing attention to the issue of fibro-fatty replacement. The description and underlying mechanisms of this phenomenon, as an element of obesity-related cardiomyopathy, have been incorporated in lines 318–327. In response to your comments, we have also discussed fibro-fatty replacement within a broader context, recognising it as a process occurring in arrhythmogenic right ventricular cardiomyopathy (page 9, paragraph 2). We further emphasised the roles of ZBTB11 and EPAS1 as factors implicated in the molecular basis of ARVC, which, in our view, merit further investigation in relation to fibro-fatty replacement observed in obesity-related cardiomyopathy, as a potential shared component in the pathogenesis of both conditions (page 9, paragraph 3, lines 16-23 and page 10, paragraph 1).

2.) Therefore, I suggest to discuss also similarities and differences between this fat-replacement in obesity-related cardiomyopathy and ARVC (which is caused by mutations in genes, encoding desmosomal cell-cell adhesion proteins).

We concur that, while fibro-fatty replacement represents a common pathological feature in the pathogenesis of both obesity-related cardiomyopathy and ARVC, the distinct molecular pathogenesis of these entities necessitates highlighting their fundamentally different clinical manifestations and prognostic outcomes. In response to your observations, we have incorporated a succinct comparative summary delineating the principal similarities and distinctions between these two disease processes (page 9, paragraph 3, lines 1-16).

3.) It would be great if the authors can increase the size of their nice figures. 

We thank you for this valuable comment. All figures and the included table have been enlarged, which we believe will improve the clarity and overall readability of the manuscript.

Reviewer 2 Report

Comments and Suggestions for Authors

This review focuses on epicardial adipose tissue as a novel therapeutic target in obesity cardiomyopathy. The review is well written with worked illustrations and tables. I suggest some minor revisions before publication:

  • Please avoid the use of abbreviations in the abstract section.
  • To visually represent the molecular and functional pathways mediating the effects of GLP-1 and GLP-1/GIP receptor agonists on EAT, a comprehensive illustration could be designed.

Author Response

1) Please avoid the use of abbreviations in the abstract section.

We are grateful for this comment. Accordingly, the abstract (page 1) has been carefully revised to eliminate all abbreviations, thereby improving its clarity and conformity with journal guidelines.

2) To visually represent the molecular and functional pathways mediating the effects of GLP-1 and GLP-1/GIP receptor agonists on EAT, a comprehensive illustration could be designed.

In response to your request, we have developed a figure (page 12, fugure 4) that succinctly depicts multiple metabolic pathways associated with epicardial adipose tissue and modulated by GLP-1 and GLP-1/GIP analogues, through which this class of agents is considered to exert cardioprotective effects. We believe that this addition provides a meaningful visual complement to the manuscript

Reviewer 3 Report

Comments and Suggestions for Authors

The study presents insight into a tissue whose functions are undervalued in clinical management and receive little attention from cardiac surgeons. The authors provide a functional narrative and the impacts of obesity and its consequent morbidity.
The detailed review of the functions and pathophysiology of lesions is well-placed, providing the reader with a comprehensive overview consistent with the evidence scattered throughout the literature. The description of the anatomical structure and its functions supports clinical discussions and offers surgeons a different perspective.
Regarding physiological modulation, the section on pharmacological modulation enriches the application of concepts during the clinical management of obese patients or those prone to obesity.
The conclusion is flawed only by failing to propose a scientific approach to clinical studies.
It was a pleasant review to read and enriched my knowledge, recalling physiological concepts and mechanisms.

Author Response

The conclusion is flawed only by failing to propose a scientific approach to clinical studies.

We concur that articulating the existing gaps in knowledge represents an essential element of any scientific discourse. Accordingly, in response to your request, we have outlined potential directions for further research on the cardioprotective effects of GLP-1 and GLP-1/GIP analogues, both at the molecular level and with respect to the macroscopic myocardial structure. We have also indicated potential methodologies that could be employed in such studies, including endomyocardial biopsy, strain assessment, and T1- and T2-weighted  magnetic resonance mapping (page 14, paragraph 2).